# LEARNING FROM SAMPLES OF VARIABLE QUALITY

**Mostafa Dehghani**
University of Amsterdam
dehghani@uva.nl

**Arash Mehrjou**
MPI for Intelligent Systems
amehrjou@tuebingen.mpg.de

**Stephan Gouws**
Google Brain
sgouws@google.com

**Jaap Kamps**
University of Amsterdam
kamps@uva.nl

**Bernhard Schölkopf**
MPI for Intelligent Systems
bs@tuebingen.mpg.de

## ABSTRACT

Training labels are expensive to obtain and may be of varying quality, as some may be from trusted expert labelers while others might be from heuristics or other sources of weak supervision such as crowd-sourcing. This creates a fundamental quality-versus-quantity trade-off in the learning process. Do we learn from the small amount of high-quality data or the potentially large amount of weakly-labeled data? We argue that if the learner could somehow know and take the label-quality into account, we could get the best of both worlds. To this end, we introduce "fidelity-weighted learning" (FWL), a semi-supervised student-teacher approach for training deep neural networks using weakly-labeled data. FWL modulates the parameter updates to a *student* network, trained on the task we care about on a per-sample basis according to the posterior confidence of its label-quality estimated by a *teacher*, who has access to limited samples with high-quality labels.

## 1 INTRODUCTION

> "*All samples are equal, but some samples are more equal than others.*"
>
> —Inspired by George Orwell quote, Animal Farm, 1945

The success of deep neural networks to date depends strongly on the availability of labeled data and usually it is much easier and cheaper to obtain small quantities of high-quality labeled data and large quantities of unlabeled data. For a large class of tasks, it is also easy to define one or more so-called "weak annotators" (Ratner et al., 2016), additional (albeit noisy) sources of *weak supervision* based on heuristics or weaker, biased classifiers trained on e.g. non-expert crowd-sourced data or data from different domains that are related. While easy and cheap to generate, it is not immediately clear if and how these additional weakly-labeled data can be used to train a stronger classifier for the task we care about.

Assuming we can obtain a large set of weakly-labeled data in addition to a much smaller training set of "strong" labels, the simplest approach is to expand the training set simply by including the weakly-supervised samples (all samples are equal). Alternatively, one may pretrain on the weak data and then fine-tune on strong data, which is one of the common practices in semi-supervised learning. We argue that treating weakly-labeled samples uniformly (i.e. each weak sample contributes equally to the final classifier) ignores potentially valuable information of the label quality (Dehghani et al., 2017c;b). Instead, we introduce Fidelity-Weighted Learning (FWL), a Bayesian semi-supervised approach that leverages a small amount of data with true labels to generate a larger training set with *confidence-weighted weakly-labeled samples*, which can then be used to modulate the fine-tuning process based on the fidelity (or quality) of each weak sample (Dehghani et al., 2018). By directly modeling the inaccuracies introduced by the weak annotator in this way, we can control the extent to which we make use of this additional source of weak supervision: more for confidently-labeled weak samples close to the true observed data, and less for uncertain samples further away from the observed data.

## 2 FIDELITY-WEIGHTED LEARNING (FWL)

In this section, we describe FWL. We assume we are given a large set of unlabeled data samples, a heuristic labeling function called the *weak annotator*, and a small set of high-quality samples labeled by experts, called the *strong dataset*, consisting of tuples of training samples $x_i$ and their true labels $y_i$, i.e. $\mathcal{D}_s = \{(x_i, y_i)\}$. We consider the latter be observations from the true target function that we are trying to learn. We use the weak annotator to generate labels for the unlabeled samples. Generated labels are noisy due to the limited accuracy of the weak annotator. This gives us the *weak dataset*

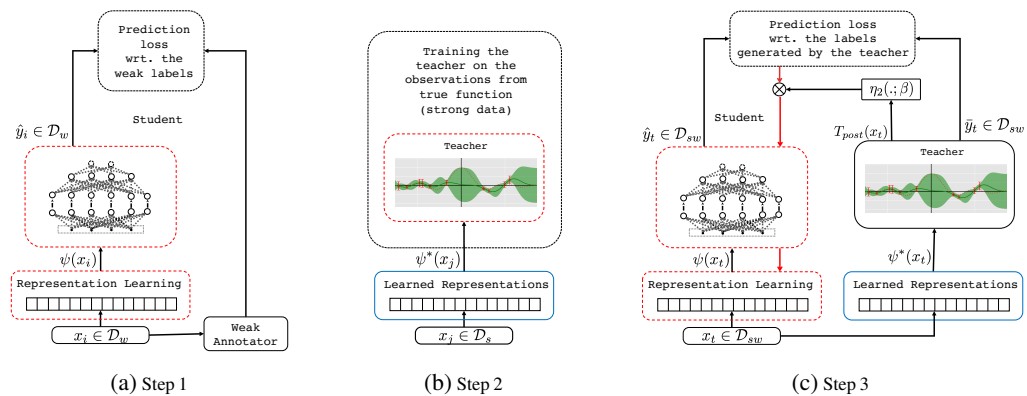

(a) Step 1          (b) Step 2          (c) Step 3

Figure 1: Illustration of Fidelity-Weighted Learning: Step 1: Pre-train student on weak data, Step 2: Fit teacher to observations from the true function, and Step 3: Fine-tune student on labels generated by teacher, taking the confidence into account. Red dotted borders and blue solid borders depict components with trainable and non-trainable parameters, respectively.

consisting of tuples of training samples $x_i$ and their weak labels $\tilde{y}_i$, i.e. $\mathcal{D}_w = \{(x_i, \tilde{y}_i)\}$. Note that we can generate a large amount of weak training data $\mathcal{D}_w$ at almost no cost using the weak annotator. In contrast, we have only a limited amount of observations from the true function, i.e. $|\mathcal{D}_s| \ll |\mathcal{D}_w|$.

Our proposed setup comprises a neural network called the **student** and a Bayesian function approximator called the **teacher**. The training process consists of three phases which we summarize in Figure 1.

**Step 1** *Pre-train the student on $\mathcal{D}_w$ using weak labels generated by the weak annotator.*

The main goal of this step is to learn a *task dependent* representation of the data as well as pretraining the student. The student function is a neural network consisting of two parts. The first part $\psi(.)$ learns the data representation and the second part $\phi(.)$ performs the prediction task (e.g. classification). Therefore the overall function is $\hat{y} = \phi(\psi(x_i))$. The student is trained on all samples of the weak dataset $\mathcal{D}_w = \{(x_i, \tilde{y}_i)\}$. For brevity, in the following, we will refer to both data sample $x_i$ and its representation $\psi(x_i)$ by $x_i$ when it is obvious from the context.

**Step 2** *Train the teacher on the strong data $(\psi(x_j), y_j) \in \mathcal{D}_s$ represented in terms of the student representation $\psi(.)$ and then use the teacher to generate a soft dataset $\mathcal{D}_{sw}$ consisting of $\langle$sample, predicted label, confidence$\rangle$ for **all** data samples.*

We use a Gaussian process as the teacher to capture the label uncertainty in terms of the student representation, estimated w.r.t the strong data. A prior mean and covariance function is chosen for $\mathcal{GP}$. The learned embedding function $\psi(\cdot)$ in Step 1 is then used to map the data samples to dense vectors as input to the $\mathcal{GP}$. We use the learned representation by the student in the previous step to compensate lack of data in $\mathcal{D}_s$ and the teacher can enjoy the learned knowledge from the large quantity of the weakly annotated data. This way, we also let the teacher see the data through the lens of the student.

The $\mathcal{GP}$ is trained on the samples from $\mathcal{D}_s$ to learn the posterior mean $\boldsymbol{m}_{\text{post}}$ (used to generate soft labels) and posterior co-variance $K_{\text{post}}(.,.)$ (which represents label uncertainty) [1]. We then create the *soft dataset* $\mathcal{D}_{sw} = \{(x_t, \bar{y}_t)\}$ using the posterior $\mathcal{GP}$, input samples $x_t$ from $\mathcal{D}_w \cup \mathcal{D}_s$, and predicted labels $\bar{y}_t$ with their associated uncertainties as computed $T(x_t) = g(\boldsymbol{m}_{\text{post}}(x_t))$ and $\Sigma(x_t) = h(K_{\text{post}}(x_t, x_t))$. The generated labels are called *soft labels*. Therefore, we refer to $\mathcal{D}_{sw}$ as a soft dataset. $g(.)$ transforms the output of $\mathcal{GP}$ to the suitable output space. For example in classification tasks, $g(.)$ would be the softmax function to produce probabilities that sum up to one. For multidimensional-output tasks where a vector of variances is provided by the $\mathcal{GP}$, the vector $K_{\text{post}}(x_t, x_t)$ is passed through an aggregating function $h(.)$ to generate a scalar value for the uncertainty of each sample. Note that we train $\mathcal{GP}$ only on the strong dataset $\mathcal{D}_s$ but then use it to generate soft labels $\bar{y}_t = T(x_t)$ and uncertainty $\Sigma(x_t)$ for samples belonging to $\mathcal{D}_{sw} = \mathcal{D}_w \cup \mathcal{D}_s$.

**Step 3** *Fine-tune the weights of the student network on the soft dataset, while modulating the magnitude of each parameter update by the corresponding teacher-confidence in its label.*

The student network of Step 1 is fine-tuned using samples from the soft dataset $\mathcal{D}_{sw} = \{(x_t, \bar{y}_t)\}$ where $\bar{y}_t = T(x_t)$. The corresponding uncertainty $\Sigma(x_t)$ of each sample is mapped to a confidence value, and this is then used to determine the step size for each iteration of the stochastic gradient descent (SGD).

---

[1] In practice, we use cluster-GP, see Appendix B.

Table 1: Descriptions of baseline models.

| | |
|---|---|
| **WA** | The weak annotator, i.e. the unsupervised method used for annotating the unlabeled data. |
| **NN$_S$** | Full Supervision Only, i.e. the student trained only on strong labeled data ($\mathcal{D}_s$). |
| **NN$_W$** | Weak Supervision Only, i.e. the or the student trained only on weakly labeled data ($\mathcal{D}_w$). |
| **NN$_{W/S}+$** | Weak Supervision + Oversampled Strong Supervision, i.e. thestudent trained on samples that are alternately drawn from $\mathcal{D}_w$ without replacement, and $\mathcal{D}_s$ with replacement. Since $|\mathcal{D}_s| \ll |\mathcal{D}_w|$, it oversamples the strong data. |
| **NN$_{W\to S}$** | Weak Supervision + Fine Tuning, i.e. the student trained on weak dataset $\mathcal{D}_w$ and fine-tuned on strong dataset $\mathcal{D}_s$. |
| **NN$_{W\omega \to NN_S}$** | The student trained on the weak data, but the step-size of each weak sample is weighted by a fixed value $0 \leq \omega \leq 1$, and fine-tuned on strong data. As an approximation for the optimal value for $\omega$, we have used the mean of $\eta_2$ of our model (below). |
| **FWL $\backslash \Sigma$** | The student trained on the weakly labeled data and fine-tuned on examples labeled by the teacher without taking the confidence into account. This baseline is similar to (Veit et al., 2017). |

So, intuitively, for data points where we have true labels, the uncertainty of the teacher is almost zero, which means we have high confidence and a large step-size for updating the parameters. However, for data points where the teacher is not confident, we down-weight the training steps of the student. This means that at these points, we keep the student function as it was trained on the weak data in Step 1.

More specifically, we update the parameters of the student by training on $\mathcal{D}_{sw}$ using SGD:

$$\boldsymbol{w}^* = \operatorname*{argmin}_{\boldsymbol{w} \in \mathcal{W}} \frac{1}{N} \sum_{(x_t, \bar{y}_t) \in \mathcal{D}_{sw}} l(\boldsymbol{w}, x_t, \bar{y}_t) + \mathcal{R}(\boldsymbol{w}),$$

$$\boldsymbol{w}_{t+1} = \boldsymbol{w}_t - \eta_t (\nabla l(\boldsymbol{w}, x_t, \bar{y}_t) + \nabla \mathcal{R}(\boldsymbol{w}))$$

where $l(\cdot)$ is the per-example loss, $\eta_t$ is the total learning rate, $N$ is the size of the soft dataset $\mathcal{D}_{sw}$, $\boldsymbol{w}$ is the parameters of the student network, and $\mathcal{R}(.)$ is the regularization term.

We define the total learning rate as $\eta_t = \eta_1(t)\eta_2(x_t)$, where $\eta_1(t)$ is the usual learning rate of our chosen optimization algorithm that anneals over training iterations, and $\eta_2(x_t)$ is a function of the label uncertainty $\Sigma(x_t)$ that is computed by the teacher for each data point. Multiplying these two terms gives us the total learning rate. In other words, $\eta_2$ represents the *fidelity* (quality) of the current sample, and is used to multiplicatively modulate $\eta_1$. Note that the first term does not necessarily depend on each data point, whereas the second term does. We propose $\eta_2(x_t) = \exp[-\beta\Sigma(x_t)]$ to exponentially decrease the learning rate for data point $x_t$ if its corresponding soft label $\bar{y}_t$ is unreliable (far from a true sample). In practice, when using mini-batches, we implement this by multiplying the loss of each example in the batch by its fidelity score and average over these fidelity-weighted losses in the batch when calculating the batch gradient based on that loss. $\beta$ is a positive scalar hyper-parameter that controls the contribution of weak and strong data to the training procedure. A small $\beta$ results in a student which listens more carefully to the teacher and copies its knowledge, while a large $\beta$ makes the student pay less attention to the teacher, staying with its initial weak knowledge. Hence, $\beta$ gives a handle to control the bias-variance trade-off. In Appendix A, we apply FWL to a one-dimensional toy problem to illustrate its various steps.

## 3  EXPERIMENTS

In this section, we apply FWL to *document ranking* task and evaluate its performance compared to the baselines presented in Table 1. Document Ranking is the core information retrieval problem and is challenging as the ranking model needs to learn a representation for long documents and capture the complex notion of relevance between queries and documents. Furthermore, the size of publicly available datasets with query-document relevance judgments is unfortunately quite small ($\sim 250$ queries). We employ a state-of-the-art pairwise neural ranker architecture as the student (Dehghani et al., 2017d) in which the ranking is cast as a regression task. Given each training sample $x$ as a triple of query $q$, and two documents $d^+$ and $d^-$, the goal is to learn a function $\mathcal{F}: \{<q, d^+, d^->\} \to \mathbb{R}$, which maps each data sample $x$ to a scalar output value $y$ indicating the probability of $d^+$ being ranked higher than $d^-$ with respect to $q$.

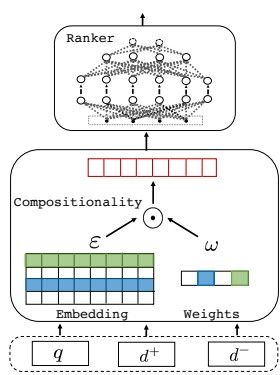

Figure 2: The student.

*The student* follows the architecture proposed in (Dehghani et al., 2017d). The first layer of the network, i.e. representation learning layer $\psi: \{<q, d^+, d^->\} \to \mathbb{R}^m$ maps each input sample to an $m$-dimensional real-valued vector. In general, besides learning embeddings for words, function $\psi$ learns to compose word embedding based on their global importance in order to generate query/document embeddings. The representation layer is followed by a simple fully-connected feed-forward network

Table 2: Performance of FWL approach and baseline methods for ranking task. $^{\blacktriangle i}$ indicates that the improvements with respect to the baseline $i$ are statistically significant at the 0.05 level using the paired two-tailed t-test with Bonferroni correction.

| Method | Robust04 | | ClueWeb | |
| --- | --- | --- | --- | --- |
| | MAP | nDCG@20 | MAP | nDCG@20 |
| $WA_{BM25}$ | $0.2503^{\blacktriangle 37}$ | $0.4102^{\blacktriangle 37}$ | $0.1021^{\blacktriangle 37}$ | $0.2070^{\blacktriangle 37}$ |
| $NN_W$ (Dehghani et al., 2017d) | $0.2702^{\blacktriangle 137}$ | $0.4290^{\blacktriangle 137}$ | $0.1297^{\blacktriangle 137}$ | $0.2201^{\blacktriangle 137}$ |
| $NN_S$ | $0.1790$ | $0.3519$ | $0.0782$ | $0.1730$ |
| $NN_{S+/W}$ | $0.2763^{\blacktriangle 1237}$ | $0.4330^{\blacktriangle 1237}$ | $0.1354^{\blacktriangle 1237}$ | $0.2319^{\blacktriangle 1237}$ |
| $NN_{W \to S}$ | $0.2810^{\blacktriangle 1237}$ | $0.4372^{\blacktriangle 1237}$ | $0.1346^{\blacktriangle 1237}$ | $0.2317^{\blacktriangle 1237}$ |
| $NN_{W\omega \to S}$ | $0.2899^{\blacktriangle 123457}$ | $0.4431^{\blacktriangle 123457}$ | $0.1320^{\blacktriangle 12347}$ | $0.2309^{\blacktriangle 12347}$ |
| $FWL \setminus \Sigma$ | $0.2980^{\blacktriangle 123457}$ | $0.4516^{\blacktriangle 123457}$ | $0.1386^{\blacktriangle 123457}$ | $0.2340^{\blacktriangle 123457}$ |
| FWL | $\mathbf{0.3124}^{\blacktriangle 12345678}$ | $\mathbf{0.4607}^{\blacktriangle 12345678}$ | $\mathbf{0.1472}^{\blacktriangle 12345678}$ | $\mathbf{0.2453}^{\blacktriangle 12345678}$ |

with a sigmoidal output unit to predict the probability of ranking $d^+$ higher than $d^-$. The general schema of the student is illustrated in Figure 2. More details are provided in Appendix C.1.

*The teacher* is implemented by clustered $\mathcal{GP}$ algorithm. See Appendix C.2 for more details. *The weak annotator* is BM25 (Robertson & Zaragoza, 2009), a well-known unsupervised method for scoring query-document pairs based on statistics of the matched terms. More details are provided in Appendix C.3. Description of the data with weak labels and data with true labels as well as the setup of the document-ranking experiments is presented in Appendix C.4 in more details.

**Results and Discussions.** We conducted k-fold cross-validation on $\mathcal{D}_s$ (the strong data) and report two standard evaluation metrics for ranking: mean average precision (MAP) of the top-ranked $1,000$ documents and normalized discounted cumulative gain calculated for the top $20$ retrieved documents (nDCG@20). Table 2 shows the performance on both datasets. As can be seen, FWL provides a significant boost on the performance over all datasets. In the ranking task, the student is designed in particular to be trained on weak annotations (Dehghani et al., 2017d), hence training the network only on weak supervision, i.e. $NN_W$ performs better than $NN_S$. This can be due to the fact that ranking is a complex task requiring many training samples, while relatively few data with true labels are available.

Alternating between strong and weak data during training, i.e. $NN_{S+/W}$ seems to bring little (but statistically significant) improvement. However, we can gain better results by the typical fine-tuning strategy, $NN_{W \to S}$. We can gain improvement by fine-tuning the $NN_W$ using labels generated by the teacher without considering their confidence score, i.e. $FWL \setminus \Sigma$. This means we just augmented the fine-tuning process by generating a fine-tuning set using teacher which is better than $\mathcal{D}_s$ in terms of quantity and $\mathcal{D}_w$ in terms of quality. This baseline is equivalent to setting $\beta = 0$. However, we see a big jump in performance when we use FWL to include the estimated label quality from the teacher, leading to the best overall results.

**Sensitivity of the FWL to the Quality of the Weak Annotator.** Our proposed setup in FWL requires defining a so-called "weak annotator" to provide a source of weak supervision for unlabelled data. In this section, we study how the quality of the weak annotator may affect the performance of the FWL on the Robust04 dataset. To do so, besides BM25 (Robertson & Zaragoza, 2009), we use three other weak annotators: vector space model (Salton & Yang, 1973) with binary term occurrence (BTO) weighting schema and vector space model with TF-IDF weighting schema, which are both weaker than BM25, and BM25+RM3 (Abdul-jaleel et al., 2004) that uses RM3 as the pseudo-relevance feedback method on top of BM25, leading to better labels.

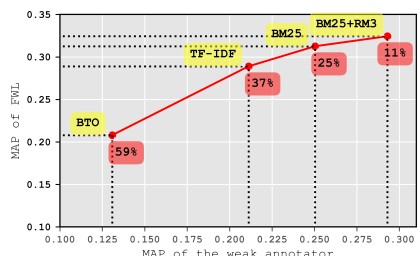

Figure 3: Performance of FWL versus performance of the corespondence weak annotator.

Figure 3 illustrates the performance of these four weak annotators in terms of their mean average precision (MAP) on the test data, versus the performance of FWL given the corresponding weak annotator. As it is expected, the performance of FWL depends on the quality of the employed weak annotator. The percentage of improvement of FWL over its corresponding weak annotator on the test data is also presented in Figure 3. As can be seen, the better the performance of the weak annotator is, the less the improvement of the FWL would be.

## 4 CONCLUSION

Training neural networks using large amounts of weakly annotated data is an attractive approach in scenarios where an adequate amount of data with true labels is not available, a situation which often arises in practice. In this paper, we introduced fidelity-weighted learning (FWL), a new student-teacher framework for semi-supervised learning in the presence of weakly labeled data. We applied FWL to document ranking and empirically verified that FWL speeds up the training process and improves over state-of-the-art semi-supervised alternatives.

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

## APPENDICES

## A  TOY EXAMPLE

To better understand FWL, we apply FWL to a one-dimensional toy problem to illustrate the various steps. Let $f_t(x) = \sin(x)$ be the true function (red dotted line in Figure 4a) from which a small set of observations $\mathcal{D}_s = \{x_j, y_j\}$ is provided (red points in Figure 4b). These observation might be noisy, in the same way that labels obtained from a human labeler could be noisy. A weak annotator function $f_w(x) = 2sinc(x)$ (magenta line in Figure 4a) is provided, as an approximation to $f_t(.)$.

The task is to obtain a good estimate of $f_t(.)$ given the set $\mathcal{D}_s$ of strong observations and the weak annotator function $f_w(.)$. We can easily obtain a large set of observations $\mathcal{D}_w = \{x_i, \tilde{y}_i\}$ from $f_w(.)$ with almost no cost (magenta points in Figure 4a).

As the teacher, we use standard Gaussian process regression[2] with this kernel:

$$k(x_i, x_j) = k_{\text{RBF}}(x_i, x_j) + k_{\text{White}}(x_i, x_j) \tag{1}$$

where,

$$k_{\text{RBF}}(x_i, x_j) = \exp\left(\frac{\|x_i - x_j\|^2}{2^2}\right)$$

$$k_{\text{White}}(x_i, x_j) = constant\_value, \quad \forall x_1 = x_2 \text{ and } 0 \text{ otherwise}$$

We fit only one $\mathcal{GP}$ on all the data points (i.e. no clustering). Also during fine-tuning, we set $\beta = 1$. The student is a simple feed-forward network with the depth of 3 layers and width of 128 neurons per layer. We have used $tanh$ as the nonlinearity for the intermediate layers and a linear output layer. As the optimizer, we used Adam (Kingma & Ba, 2015) and the initial learning rate has been set to 0.001. We randomly sample 100 data points from the weak annotator and 10 data points from the true function. We introduce a small amount of noise to the observation of the true function to model the noise in the human labeled data.

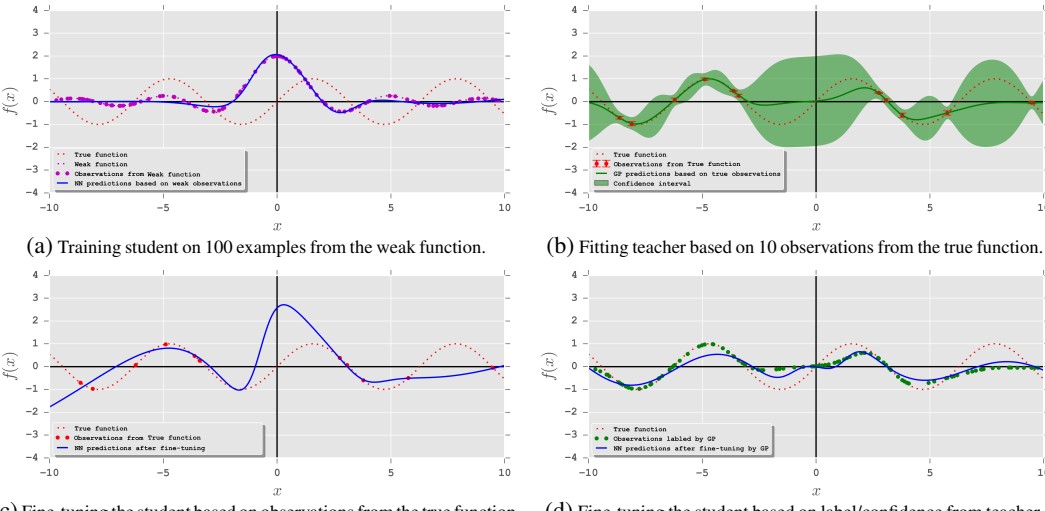

(a) Training student on 100 examples from the weak function.

(b) Fitting teacher based on 10 observations from the true function.

(c) Fine-tuning the student based on observations from the true function.

(d) Fine-tuning the student based on label/confidence from teacher.

Figure 4: Toy example: The true function we want to learn is $y = \sin(x)$ and the weak function is $y = 2sinc(x)$.

We consider two experiments:

1. A neural network trained on weak data and then fine-tuned on strong data from the true function, which is the most common semi-supervised approach (Figure 4c).
2. A teacher-student framework working by the proposed FWL approach.

As can be seen in Figure 4d, FWL by taking into account label confidence, gives a better approximation of the true hidden function. We repeated the above experiment 10 times. The average RMSE with respect to the true function on a set of test points over those 10 experiments for the student, were as follows:

---

[2]http://gpflow.readthedocs.io/en/latest/notebooks/regression.html

---

**Algorithm 1** Clustered Gaussian processes.

---

1: Let $N$ be the sample size, $n$ the sample size of each cluster, $K$ the number of clusters, and $c_i$ the center of cluster $i$.
2: Run K-means with $K$ clusters over all samples with true labels $\mathcal{D}_s = \{x_i, y_i\}$.

$$\text{K-means}(x_i) \to c_1, c_2, ..., c_K$$

where $c_i$ represents the center of cluster $C_i$ containing samples $D_s^{c_i} = \{x_{i,1}, x_{i,2}, ... x_{i,n}\}$.
3: Assign each of $K$ clusters a Gaussian process and train them in parallel to approximate the label of each sample.

$$
\begin{aligned}
\mathcal{GP}_{c_i}(\boldsymbol{m}_{\text{post}}^{c_i}, K_{\text{post}}^{c_i}) &= \mathcal{GP}(\boldsymbol{m}_{\text{prior}}, K_{\text{prior}}) | D_s^{c_i} = \{(\psi(x_{s,c_i}), y_{s,c_i})\} \\
T_{c_i}(x_t) &= g(\boldsymbol{m}_{\text{post}}^{c_i}(x_t)) \\
\Sigma_{c_i}(x_t) &= h(K_{\text{post}}^{c_i}(x_t, x_t))
\end{aligned}
$$

where $\mathcal{GP}_{c_i}$ is trained on $\mathcal{D}_s^{c_i}$ containing samples belonging to the cluster $c_i$. Other elements are defined in Section 2
4: Use trained teacher $T_{c_i}(.)$ to evaluate the soft label and uncertainty for samples from $\mathcal{D}_{sw}$ to compute $\eta_2(x_t)$ required for step 3 of FWL. We use $T(.)$ as a wrapper for all teachers $\{T_{c_i}\}$.

---

1. Student is trained on weak data (blue line in Figure 4a): 0.8406,
2. Student is trained on weak data then fine tuned on true observations (blue line in Figure 4c): 0.5451,
3. Student is trained on weak data, then fine tuned by soft labels and confidence information provided by the teacher (blue line in Figure 4d): 0.4143 (best).

## B  DETAILED DESCRIPTION OF CLUSTERED GP

We suggest using several $\mathcal{GP} = \{GP_{c_i}\}$ to explore the entire data space more effectively. Even though inducing points and stochastic methods make $\mathcal{GP}$s more scalable we still observed poor performance when the entire dataset was modeled by a single $\mathcal{GP}$. Therefore, the reason for using multiple $\mathcal{GP}$s is mainly empirically inspired by (Shen et al., 2006) which is explained in the following:
We used the Sparse Gaussian Process implemented in GPflow. The algorithm is scalable in the sense that it is not $O(N^3)$ as original $\mathcal{GP}$ is. It introduces inducing points in the data space and defines a variational lower bound for the marginal likelihood. The variational bound can now be optimized by stochastic methods which make the algorithm applicable in large datasets. However, the tightness of the bound depends on the location of inducing points which are found through the optimization process. The pseudo-code of the clustered $\mathcal{GP}$ is presented in Algorithm 1. When the main issue is computational resources (when the number of inducing points for each $\mathcal{GP}$ is large), we can first choose the number $n$ which is the maximum size of the dataset on which our resources allow to train a $\mathcal{GP}$, then find the number of clusters $K = N/n$ accordingly. The rest of the algorithm remains unchanged.

## C  DETAILED SETUP OF THE MODEL AND EXPERIMENTS

### C.1  DETAILED ARCHITECTURE OF THE STUDENTS

The employed student is proposed in (Dehghani et al., 2017d). The first layer of the network models function $\psi$ that learns the representation of the input data samples, i.e. $(q, d^+, d^-)$, and consists of three components: (1) an embedding function $\varepsilon : \mathcal{V} \to \mathbb{R}^m$ (where $\mathcal{V}$ denotes the vocabulary set and $m$ is the number of embedding dimensions), (2) a weighting function $\omega : \mathcal{V} \to \mathbb{R}$, and (3) a compositionality function $\odot : (\mathbb{R}^m, \mathbb{R})^n \to \mathbb{R}^m$. More formally, the function $\psi$ is defined as:

$$
\begin{aligned}
\psi(q, d^+, d^-) = [&\odot_{i=1}^{|q|}(\varepsilon(t_i^q), \omega(t_i^q)) \, || \\
&\odot_{i=1}^{|d^+|}(\varepsilon(t_i^{d^+}), \omega(t_i^{d^+})) \, || \\
&\odot_{i=1}^{|d^-|}(\varepsilon(t_i^{d^-}), \omega(t_i^{d^-}))],
\end{aligned}
\tag{2}
$$

where $t_i^q$ and $t_i^d$ denote the $i^{th}$ term in query $q$ respectively document $d$. The embedding function $\varepsilon$ maps each term to a dense $m$- dimensional real value vector, which is learned during the training phase. The weighting function $\omega$ assigns a weight to each term in the vocabulary. It has been shown that $\omega$ simulates the effect of inverse document frequency (IDF), which is an important feature in information retrieval (Dehghani et al., 2017d).

The compositionality function $\odot$ projects a set of $n$ embedding-weighting pairs to an $m$- dimensional representation, independent from the value of $n$:

$$\bigodot_{i=1}^{n}(\varepsilon(t_i),\omega(t_i)) = \frac{\sum_{i=1}^{n}\exp(\omega(t_i))\cdot\varepsilon(t_i)}{\sum_{j=1}^{n}\exp(\omega(t_j))}, \tag{3}$$

which is in fact the normalized weighted element-wise summation of the terms' embedding vectors. Again, it has been shown that having global term weighting function along with embedding function improves the performance of ranking as it simulates the effect of inverse document frequency (IDF). In our experiments, we initialize the embedding function $\varepsilon$ with word2vec embeddings (Mikolov et al., 2013) pre-trained on Google News and the weighting function $\omega$ with IDF.

The representation layer is followed by a simple fully connected feed-forward network with $l$ hidden layers followed by a softmax which receives the vector representation of the inputs processed by the representation learning layer and outputs a prediction $\tilde{y}$. Each hidden layer $z_k$ in this network computes $z_k = \alpha(W_k z_{k-1} + b_k)$, where $W_k$ and $b_k$ denote the weight matrix and the bias term corresponding to the $k^{th}$ hidden layer and $\alpha(.)$ is the non-linearity. These layers follow a sigmoid output. We employ the cross entropy loss:

$$\mathcal{L}_t = \sum_{i\in B}[-y_i\log(\hat{y}_i) - (1-y_i)\log(1-\hat{y}_i)], \tag{4}$$

where $B$ is a batch of data samples.

## C.2 DETAILED ARCHITECTURE OF THE TEACHERS

We use Gaussian Process as the teacher and pass the mean of $\mathcal{GP}$ through the same function $g(.)$ that is applied on the output of the student network. $h(.)$ is an aggregation function that takes variance over several dimensions and outputs a single measure of variance. As a reasonable choice, the aggregating function $h(.)$ in our sentiment classification task (three classes) is *mean* of variances over dimensions. In the teacher, linear combinations of different kernels are used in our experiments.

We use sparse variational GP regression[3] (Titsias, 2009) with this kernel:

$$k(x_i, x_j) = k_{\text{Matern3/2}}(x_i, x_j) + k_{\text{Linear}}(x_i, x_j) + k_{\text{White}}(x_i, x_j) \tag{5}$$

where,

$$k_{\text{Matern3/2}}(x_i, x_j) = \left(1 + \frac{\sqrt{3}\|x_i - x_j\|}{l}\right)\exp\left(-\frac{\sqrt{3}\|x_i - x_j\|}{l}\right)$$
$$k_{\text{Linear}}(x_i, x_j) = \sigma_0^2 + x_i.x_j$$
$$k_{\text{White}}(x_i, x_j) = constant\_value, \quad \forall x_1 = x_2 \text{ and } 0 \text{ otherwise}$$

We empirically found $l = 1$ satisfying value for the length scale of Matern3/2 kernels. We also set $\sigma_0 = 0$ to obtain a homogeneous linear kernel. The constant value of $K_{White}(.,.)$ determines the level of noise in the labels. This is different from the noise in weak labels. This term explains the fact that even in true labels there might be a trace of noise due to the inaccuracy of human labelers. We set the number of clusters in the clustered $\mathcal{GP}$ algorithm for the ranking task to $50$.

## C.3 WEAK ANNOTATORS

The weak annotator is BM25 (Robertson & Zaragoza, 2009), a well-known unsupervised retrieval method. This method heuristically scores a given pair of query-document based on the statistics of their matched terms. In the pairwise document ranking setup, $\tilde{y}_i$ for a given sample $x_j = (q, d^+, d^-)$ is the probability of document $d^+$ being ranked higher than $d^-$: $\tilde{y}_i = P_{q,d^+,d^-} = s_{q,d^+}/s_{q,d^+} + s_{q,d^-}$, where $s_{q,d}$ is the score obtained from the weak annotator.

## C.4 DATA COLLECTION, PARAMETERS AND SETUP

**Collections** We use two standard TREC collections for the task of ad-hoc retrieval: The first collection (*Robust04*) consists of 500k news articles from different news agencies as a homogeneous collection. The second collection

---

[3]http://gpflow.readthedocs.io/en/latest/notebooks/SGPR_notes.html

(*ClueWeb*) is ClueWeb09 Category B, a large-scale web collection with over 50 million English documents, which is considered as a heterogeneous collection. Spam documents were filtered out using the Waterloo spam scorer [4] (Cormack et al., 2011) with the default threshold 70%.

**Data with true labels** We take query sets that contain human-labeled judgments: a set of 250 queries (TREC topics 301–450 and 601–700) for the Robust04 collection and a set of 200 queries (topics 1-200) for the experiments on the ClueWeb collection. For each query, we take all documents judged as relevant plus the same number of documents judged as non-relevant and form pairwise combinations among them.

**Data with weak labels** We create a query set $Q$ using the unique queries appearing in the AOL query logs (Pass et al., 2006). This query set contains web queries initiated by real users in the AOL search engine that were sampled from a three-month period from March 2006 to May 2006. We applied standard pre-processing Dehghani et al. (2017d;a) on the queries: We filtered out a large volume of navigational queries containing URL substrings ("http", "www.", ".com", ".net", ".org", ".edu"). We also removed all non-alphanumeric characters from the queries. For each dataset, we took queries that have at least ten hits in the target corpus using our weak annotator method. Applying all these steps, We collect 6.15 million queries to train on in Robust04 and 6.87 million queries for ClueWeb. To prepare the weakly labeled training set $\mathcal{D}_w$, we take the top 1,000 retrieved documents using BM25 for each query from training query set $Q$, which in total leads to $\sim |Q| \times 10^6$ training samples.

**Setup** For the evaluation of the whole model, we conducted 3-fold cross-validation. However, for each dataset, we first tuned all the hyper-parameters of the student in the first step on the set with true labels using batched GP bandits with an expected improvement acquisition function (Desautels et al., 2014) and kept the optimal parameters of the student fixed for all the other experiments. The size and number of hidden layers for the student is selected from $\{64, 128, 256, 512\}$. The initial learning rate and the dropout parameter were selected from $\{10^{-3}, 10^{-5}\}$ and $\{0.0, 0.2, 0.5\}$, respectively. We considered embedding sizes of $\{300, 500\}$. The batch size in our experiments was set to 128. We use ReLU (Nair & Hinton, 2010) as a non-linear activation function $\alpha$ in student. We use the Adam optimizer (Kingma & Ba, 2015) for training, and *dropout* (Srivastava et al., 2014) as a regularization technique.

At inference time, for each query, we take the top 2,000 retrieved documents using BM25 as candidate documents and re-rank them using the trained models. We use the Indri[5] implementation of BM25 with default parameters (i.e., $k_1 = 1.2$, $b = 0.75$, and $k_3 = 1,000$).

---

[4] http://plg.uwaterloo.ca/~gvcormac/clueweb09spam/
[5] https://www.lemurproject.org/indri.php

