# OpenReview forum: "Learning from Samples of Variable Quality"
_ICLR.cc/2019/Workshop/LLD — LLD 2019_

### Official Review · AnonReviewer2 · 2019-04-08
**interesting setup, but methods not well motivated and explained**

**Rating:** 2
**Confidence:** 2

**Review:**

The authors propose a distillation approach to learn from weakly annotated data aside from a strongly annotated data. The general setup and approach of distillation seems interesting, but it is unclear what the resulted model will learn to do. Specifically, I have the following concerns/questions:

- the proposed method is kind of hacky, in the sense that it is not well motivated and grounded on any specific intuition, theoretical or not.
- it is not clear to me what the student will converge to after training in the step 3. will it just learn to predict whatever the teacher outputs? if that's the case why do we distill it, aside from the obvious fact that now it is parametric rather than a GP?
- following the previous point, have you compared the performance of the student with the performance of the teacher?

---

### Official Review · AnonReviewer1 · 2019-04-08
**Review of "Learning from Samples of Variable Quality"**

**Rating:** 4
**Confidence:** 2

**Review:**

Summary of the paper:
This paper proposes a technique to tackle the case where the labels of a supervised problem come from different sources and have varying quality. To this end, a three steps approach called FWL (Fidelity Weighted Learning) is proposed. In particular, FWL consists in constructing a “student” and a “teacher” interacting to refine the quality of the prediction.

Some numerical experiments supporting the approach are proposed. In particular, the FWL algorithm is tested on the problem of document ranking.
An investigation of the sensitivity of the approach to the quality of the weak annotator is also proposed.


A few comments and questions:
-the fact that the algorithm uses a Neural Net should be emphasized more in the introduction
-Can the proposed approach be extended to other learning algorithm (i.e. not only a NN)?
-Why specifically use a Gaussian process? Can another approach be used?
-concerning the investigation on the “sensitivity of the FWL to the quality of the weak annotator” (sic),
does a “phase transition” occur? i.e. is there a threshold on the quality of the weak annotator below which FWL does not yield any improvement?

Reviewer’s assessment:
I found the paper to be fairly well written. The ideas are exposed clearly and the numerical results support the approach. Since the topic of this work clearly falls within the scope of the workshop, I recommend to accept this paper.

---

### Decision · Program_Chairs · 2019-04-16
**Acceptance Decision**

Accept